# HoVer-NeXt: A Fast Nuclei Segmentation and Classification Pipeline for Next Generation Histopathology

**Elias Baumann**[1]                                           ELIAS.BAUMANN@UNIBE.CH
**Bastian Dislich**[1]
**Josef Lorenz Rumberger**[2,3]                  JOSEFLORENZ.RUMBERGER@MDC-BERLIN.DE
**Iris D. Nagtegaal**[4]
**María Rodríguez Martínez**[5]
**Inti Zlobec**[1]

[1] *Institute of Tissue Medicine and Pathology, University of Bern, Bern, Switzerland*

[2] *Max-Delbrück-Center for Molecular Medicine in the Helmholtz Association, Berlin, Germany*

[3] *Humboldt-Universität zu Berlin, Faculty of Mathematics and Natural Sciences, Berlin, Germany*

[4] *Department of Pathology, Radboud Institute for Molecular Life Sciences, Radboud University Medical Center, Nijmegen, The Netherlands*

[5] *Yale School of Medicine*

**Editors:** Accepted for publication at MIDL 2024

## Abstract

In cancer, a variety of cell types, along with their local density and spatial organization within tissues, play a key role in driving cancer progression and modulating patient outcomes. At the basis of cancer diagnosis is the histopathological assessment of tissues, stained by hematoxylin & eosin (H&E), which gives the nuclei of cells a dark purple appearance, making them particularly distinguishable and quantifiable. The identification of individual nuclei, whether in a proliferating (mitosis) or resting state, and their further phenotyping (e.g. immune cells) is the foundation on which histopathology images can be used for further investigations into cellular interaction, prognosis or response prediction. To this end, we develop a H&E based nuclei segmentation and classification model that is both fast (1.8s/mm2 at 0.5mpp, 3.2s/mm2 at 0.25mpp) and accurate (0.84 binary F1, 0.758 mean balanced Accuracy) which allows us to investigate the cellular composition of large-scale colorectal cancer (CRC) cohorts. We extend the publicly available Lizard CRC nuclei dataset with a mitosis class and publish further validation data for the rarest classes: mitosis and eosinophils. Moreover, our pipeline is 5× faster than the CellViT pipeline, 17× faster than the HoVer-Net pipeline, and performs competitively on the PanNuke pan-cancer nuclei dataset (47.7 mPQ$_{Tiss}$, +3% over HoVer-Net). Our work paves the way towards extensive single-cell information directly from H&E slides, leading to a quantitative view of whole slide images. Code, model weights as well as all additional training and validation data, are publicly available on github.

**Keywords:** Panoptic segmentation, Nuclei segmentation and classification, Deep Learning, Histopathology, Colorectal Cancer

## 1. Introduction

Histopathological assessment of tissues is a cornerstone for the diagnosis and prognosis of diseases, including cancer. Among the most frequent and deadly is colorectal cancer (CRC), for which the overall 5-year survival rate is only around 65% (Siegel et al., 2023). Tissue biomarkers play a crucial role in improving prognostication and designing more personalized treatments for individual patients. In recent years, deep neural networks have

shown promising results in biomarker prediction (Kather et al., 2019; Bulten et al., 2019) and discovery (Zheng et al., 2023), as well as other tasks such as segmentation and classification of biological structures (Ronneberger et al., 2015; Havaei et al., 2015). One of the emerging tasks is the identification, classification, and segmentation (i.e. panoptic segmentation) of cells or more commonly their nuclei directly on routine diagnostic slides (Gamper et al., 2019; Graham et al., 2021a). Most cell types can only be differentiated on H&E images at high magnifications (i.e. 20x or 40x), because identification is based on nuclear morphology, texture and tissue context. However, Whole Slide Images (WSI) at high magnification are large with sizes above $100000 \times 100000$ pixels, and inference runtimes of currently available models make it infeasible to run them for clinical routine or investigations on large-scale cohorts. Methods for panoptic segmentation include HoVer-Net (Graham et al., 2018), which uses an encoder-decoder architecture with three decoders, one for semantic and two for watershed-based instance segmentation. In comparison, the panoptic segmentation version of StarDist (Weigert and Schmidt, 2022) has a similar architecture, but only one decoder for instance segmentation which predicts star-shaped polygon mask proposals for the individual cells and post-processes them to instances by means of a non-maximum suppression algorithm. CellViT (Hörst et al., 2023) then further improves state-of-the-art by using a SAM (Kirillov et al., 2023) encoder combined with HoVer-Nets decoders. On the other hand, Tommasino et al. (2023) propose a simplified HoVer-UNet for $3\times$ speedup over HoVer-Net. Recently, the CoNiC challenge tried to find new best practices in nuclei segmentation and classification (Graham et al., 2023, 2021b). They found that the top three submissions, which included our own, are based on newer encoders such as EfficientNet-v2 (Tan and Le, 2021), and tackled class imbalance via class distribution-based importance sampling and loss weighting (Weigert and Schmidt, 2022; Rumberger et al., 2022; Zhang and Zhang, 2022). The CoNiC challenge dataset (Lizard) (Graham et al., 2021a), which is based on H&E CRC images at ∼0.5mpp, includes six classes: lymphocytes, neutrophils, plasma cells, eosinophils, epithelial cells and connective-tissue cells. The post-challenge analysis Graham et al. (2023), demonstrates the value of the dataset and such methods by successfully applying them to tumor grading and patient survival prediction tasks. However, Lizard does not consider mitoses as separate objects of interest. Rather, they are classified as epithelium, lymphocyte, neutrophil, or not at all (c.f. Figure 1 D). Mitoses are indicators of the proliferative activity of tumors and have an impact on treatment decisions for e.g. breast or pancreatic neuroendocrine tumors (Łukasiewicz et al., 2021; Kim and Hong, 2016). In Summary, long inference times of publicly available models, incompatible with large-scale investigations, coupled with the absence of biologically relevant class annotations such as mitosis, motivate the subsequent contributions:

1. We developed HoVer-NeXt (HN), an updated model based on our CoNiC challenge submission, which retains high performance on the Lizard cell types while also predicting mitoses.

2. We provide an additional mitosis training dataset, modify Lizard to include mitosis, and publish both together with additional validation sets for mitoses and eosinophils.

3. We provide a model trained on the PanNuke (Gamper et al., 2019) pan-cancer panoptic segmentation dataset, which shows competitive performance.

4. We construct a highly efficient WSI inference pipeline, which achieves a 17× speedup over the HoVer-Net pipeline and a 5× speedup over the CellViT pipeline on whole slide inference.

Code for training, inference, weights for all models as well as links to data can be found here: github.com/digitalpathologybern/hover_next_train, and here /hover_next_inference.

## 2. Methods

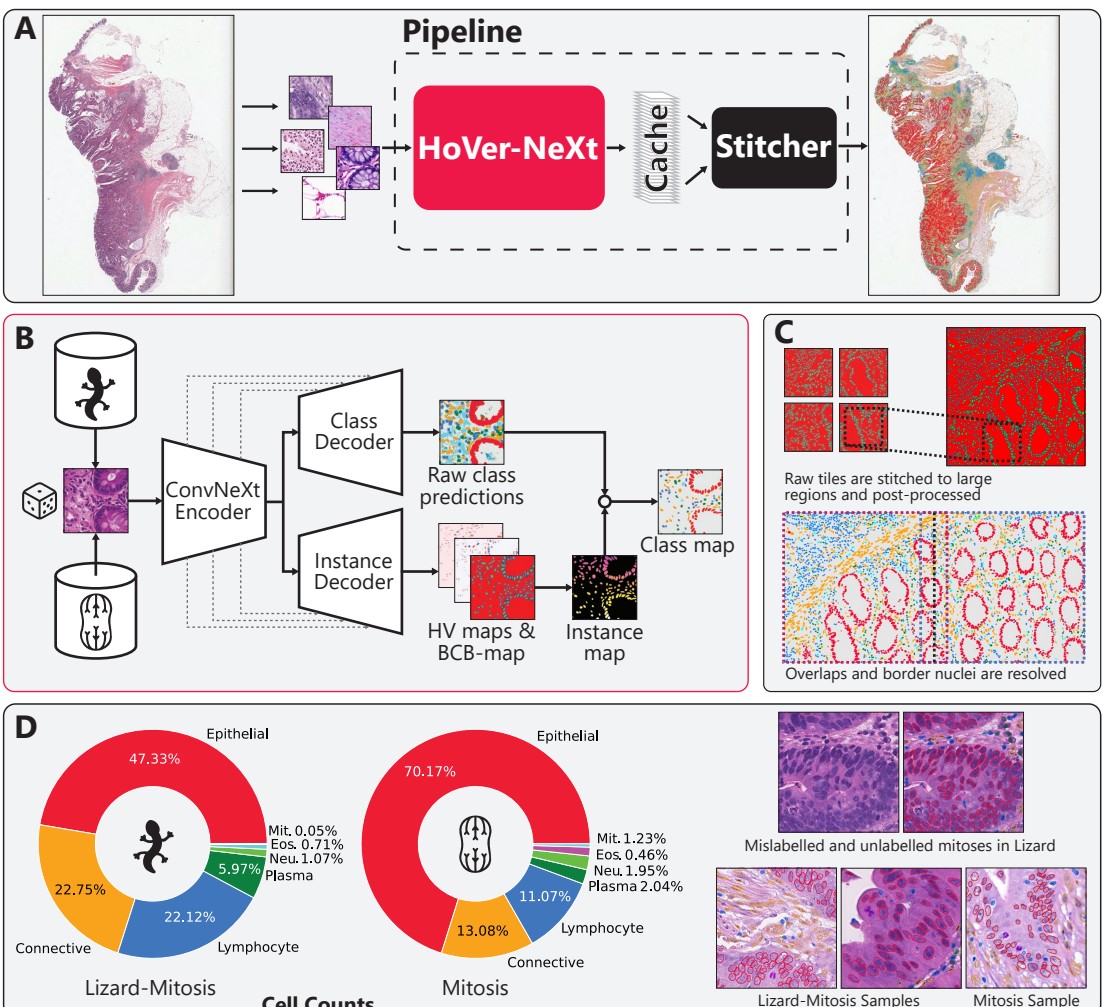

Figure 1: Our proposed pipeline consists of the Model (HoVer-NeXt) and a separate Stitcher that post-processes the output(A). HoVer-NeXt is trained with random sampling from Lizard-Mitosis and Mitosis, and uses a U-Net architecture with two decoders to produce raw class predictions, center-point vectors and a boundary, nuclei center and background map(B). For fast inference, tiles are pre-stitched, then post-processed and then overlaps are resolved(C). Lizard-Mitosis and Mitosis have differing distributions and strong class imbalance (D).

## 2.1. Summary of the CoNiC 2022 Submission

Starting from the HoVer-Net model setup, we propose several simplifications and extensions. Firstly, we combine the two instance segmentation decoders into a single decoder. The binary nuclei segmentation map is replaced by a 3-class nuclei boundary, nuclei center, background prediction map (BCB-map), which showed improved results in other modalities (Caicedo et al., 2018), and can directly be used for watershed-based instance segmentation, reducing the need for additional post-processing steps. The HoVet-Net HV maps or center-point vectors are thus only used as an auxiliary task (Hirsch and Kainmueller, 2020). As the architecture, we use a U-Net (Ronneberger et al., 2015) with an EfficientNet-V2 encoder (Tan and Le, 2021). To tackle the class imbalance in Lizard, we employ class-based importance sampling using per-pixel class statistics as weights and use focal loss with class weighting proportional to the inverse of the exponential moving average class prior (Araslanov and Roth, 2021). Finally, model outputs are post-processed with class-specific hyperparameters, as the classes differ in average object size and shape. For more details we refer to our previous publication for the CoNiC Challenge (Rumberger et al., 2022). While the setup and processing steps were feasible within the scope of optimizing for the challenge metrics, they do not scale well to WSI. Therefore, we optimize the model and embed it in a pipeline for efficient WSI inference.

## 2.2. HoVer-NeXt

We updated the model with a ConvNeXt-v2 (Woo et al., 2023) encoder, which shows competitive results on a variety of benchmarks (Roy et al., 2023). ConvNeXt-v2 uses a larger pooling operation which we accommodate for by adding an additional upsampling step to maintain the same U-Net depth. In our experiments, we use ConvNeXt-v2 Tiny, Base and Large. We further simplify the model by replacing class based loss weighting with a standard focal loss (Lin et al., 2017), since data sampling is already sufficient to treat the label imbalance (Ablation: See Supp. C.2). A convex-hull-based post-processing step is also removed as individual convex hull computations are computationally expensive. Tile-based normalization leads to artifacts in out-of-distribution tiles and is removed as well (See Supp. Figure 6). The training setup can be found in Supp. A.1. Beyond these changes, we setup an easy-to-use WSI inference pipeline.

## 2.3. Inference Pipeline

Relevant foreground area on the WSI is first identified using a threshold on the gray scale representation of the WSI thumbnail (Details: Supp. A.3). The model then processes tiles with overlap (8px/0.5mpp, 16px/0.25mpp) with classmap and BCB-map being compressed and stored on disk. For test-time-augmentations (TTAs), we only include HED color augmentation, mirroring and 90° rotation to avoid negative effects of augmentations (Details: Supp. A.4, Augmentation Parameters: Supp. C.1). Tiles are center-cropped and stitched to large regions for parallel processing. Then, based on individual class thresholds, foreground area and seed points are generated in the BCB map and then processed with a watershed algorithm to get nuclei instances. Small holes in instances are removed and false merges are resolved. Classes are assigned based on majority vote and instances are filtered based on class-specific size thresholds, which are determined via hyperparameter search on the validation set. Finally, overlaps between large ROIs are resolved (Details: Supp. A.2).

**Optimizing for speed:** To further optimize the inference time, storage, and memory requirements of the pipeline, the model runs at half-precision and the class output is quantized by mapping softmax outputs to values between 0-255. Outputs are written to disk without post-processing by a separate process to ensure high GPU utilization. Data is stored as LZ4 compressed Zarr arrays, allowing for fast compression and concurrent writes and reads. Finally, the pre-stitching of raw inference tiles to large regions allows us to avoid resolving large numbers of overlaps, retain the option to parallelize watershed, and keep the peak memory consumption low.

## 2.4. Datasets

**Lizard and PanNuke** Lizard and PanNuke are publicly available H&E panoptic nuclei segmentation datasets, one CRC specific and one pan-cancer (More details: Supp. A.5). To be able to compare HoVer-NeXt to our own challenge submission, we use the same 80% train, 20% validation split with the "GlaS" subset as an out-of-distribution test set. Conclusions drawn in the challenge evaluation (Graham et al., 2023) are therefore likely to be translatable to our new models. To compare our model with HoVer-UNet (Tommasino et al., 2023), CellViT (Hörst et al., 2023) as well as HoVer-Net (Graham et al., 2018), we also include the PanNuke dataset as a benchmark.

**Mitosis and Lizard-Mitosis** We create our own dataset specifically for mitoses and extend the Lizard dataset with mitoses. To achieve this, we select 48 ROIs ($8192 \times 8192$px) from 11 H&E stained CRC WSI, create a mitosis specific pHH3 immunohistochemistry restain, register the images and generate ground truth by thresholds on the stain deconvolved DAB channel (see Supp. A.6). To generate panoptic segmentation labels for this dataset beyond mitoses, as well as re-labeling Lizard for mitoses, we adapt a self-training routine proposed by Yang et al. (2021) (see Supp. A.7).

**Further validation: MitEval and EosEval** Additionally, we create two holdout test sets, one with eosinophils manually annotated by a board-certified pathologist in 11 ROIs of CRC resection WSI respectively, and one with 3 board-certified pathologists annotating 13 ROIs for mitosis (Supp. A.8). For both datasets, we report WSI-level performance.

## 2.5. Evaluation Metrics

Foucart et al. (2023) show that panoptic quality should be avoided for the evaluation of nuclei segmentation and classification. Panoptic quality is defined as the product of the detection F1 score and Intersection over Union (IoU) of true positives. However, the small size of nuclei makes IoU too sensitive for coarse annotations. Moreover, the aggregation of IoU and F1 score incentivizes not detecting an instance at all over misclassifying it. We therefore employ their guidelines, yet also report panoptic quality for comparison. For binary segmentation, we use F1-Score and Matthews correlation coefficient (MCC). For detection, we use distance-based matching ($6\mu$m@0.5mpp, $12\mu$m@0.25mpp (Sirinukunwattana et al., 2016)) and evaluate the detections using balanced accuracy and F1 Score. Detection metrics for Lizard are evaluated on $248 \times 248$ center crops to avoid having to detect nuclei with their center outside of the tile. For segmentation, we use Hausdorff distance, which has the advantage of also considering shape irregularities that IoU would miss (Foucart et al., 2023).

## 3. Results

### 3.1. Lizard-Mitosis

**A** Balanced Accuracy

| | mAcc | Neu | Epi | Lym | Pla | Eos | Con |
|---|---|---|---|---|---|---|---|
| HN$_{CoNiC}$ | 0.748 | **0.650** | 0.789 | 0.840 | **0.706** | 0.711 | 0.794 |
| HN$_{Large}$ | **0.759** | 0.627 | **0.796** | 0.858 | 0.693 | **0.785** | 0.794 |
| HN$_{Base}$ | 0.752 | 0.613 | 0.791 | **0.867** | 0.681 | 0.763 | 0.794 |
| HN$_{Tiny}$ | 0.749 | 0.623 | 0.789 | 0.851 | 0.672 | 0.775 | 0.786 |

Hausdorff Distance

| | Neu | Epi | Lym | Pla | Eos | Con |
|---|---|---|---|---|---|---|
| HN$_{CoNiC}$ | **1.966** | **2.722** | **1.145** | **1.103** | 2.176 | **1.976** |
| HN$_{Large}$ | 2.250 | 2.843 | 1.165 | 1.196 | 2.149 | 2.003 |
| HN$_{Base}$ | 2.533 | 2.857 | 1.206 | 1.229 | **2.134** | 2.021 |
| HN$_{Tiny}$ | 2.663 | 2.899 | 1.197 | 1.269 | 2.432 | 2.051 |

F1 Score

| | bF1* | mF1 | Neu | Epi | Lym | Pla | Eos | Con |
|---|---|---|---|---|---|---|---|---|
| HN$_{CoNiC}$ | 0.832 | 0.600 | 0.300 | 0.822 | 0.759 | **0.505** | 0.516 | 0.701 |
| HN$_{Large}$ | **0.841** | **0.606** | **0.313** | **0.826** | 0.766 | 0.471 | 0.553 | 0.708 |
| HN$_{Base}$ | 0.841 | 0.594 | 0.261 | 0.822 | **0.767** | 0.451 | **0.557** | **0.708** |
| HN$_{Tiny}$ | 0.836 | 0.572 | 0.205 | 0.822 | 0.752 | 0.432 | 0.522 | 0.698 |

Panoptic Quality

| | bPQ* | mPQ | Neu | Epi | Lym | Pla | Eos | Con |
|---|---|---|---|---|---|---|---|---|
| HN$_{CoNiC}$ | **0.546** | 0.454 | 0.196 | **0.615** | 0.636 | **0.423** | 0.342 | 0.513 |
| HN$_{Large}$ | 0.546 | **0.454** | **0.206** | 0.606 | **0.644** | 0.386 | 0.369 | **0.516** |
| HN$_{Base}$ | 0.539 | 0.440 | 0.150 | 0.602 | 0.637 | 0.366 | **0.373** | 0.512 |
| HN$_{Tiny}$ | 0.534 | 0.420 | 0.121 | 0.598 | 0.624 | 0.348 | 0.324 | 0.506 |

**B** Binary Pixel Metrics

| | HN$_{CoNiC}$ | HN$_{Large}$ | HN$_{Base}$ | HN$_{Tiny}$ |
|---|---|---|---|---|
| F1 | 0.814 | **0.819** | 0.818 | 0.814 |
| MCC | 0.776 | **0.786** | 0.785 | 0.781 |

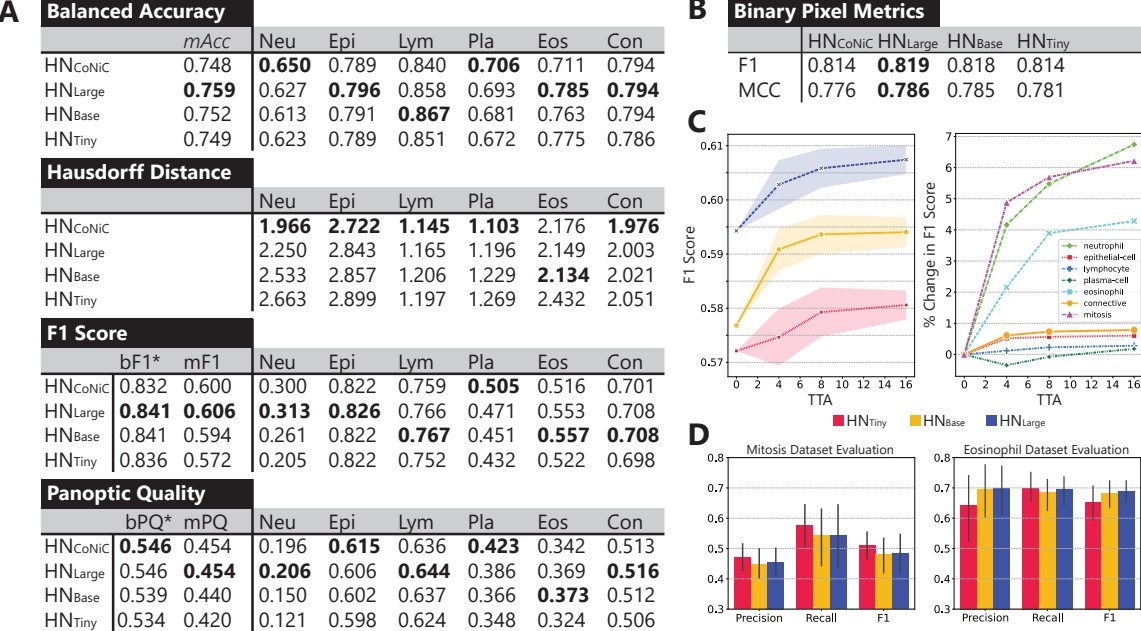

Figure 2: Results on the GlaS test-set, including comparison against HN$_{CoNiC}$ for detection and segmentation metrics (A), binary segmentation (B), and F1 Score differences when using varying numbers of test-time augmentations (C). Detection results on EosVal and MitVal (D)

In the first experiment we compare our new HoVer-NeXt model with our CoNiC submission (referred to as HN$_{CoNiC}$) on the GlaS test set. All results shown in Figure 2 are obtained with 16TTA. For binary segmentation, all three HoVer-NeXt models achieve higher F1 score and MCC, with HN$_{Large}$ having +0.005 F1-Score HN$_{CoNiC}$ and HN$_{Tiny}$ being on par with HN$_{CoNiC}$ with 0.814 binary pixel F1 score. These improvements are reflected in an increase of +0.009 for HN$_{Large}$ over the 0.832 baseline for binary detection F1. Considering class-specific classification, the results are more diverse, with HN$_{CoNiC}$ achieving the highest balanced accuracy in neutrophil (0.706) and plasma cell (0.65) classification. HN$_{Base}$ achieves the highest balanced accuracy on lymphocytes (0.867) and is on par with HN$_{Large}$ and HN$_{CoNiC}$ on connective tissue cells (0.794). On eosinophils, we observe the largest change with HN$_{Large}$ at 0.785 (+0.074) and even HN$_{Tiny}$ (+0.064) largely increases in accuracy over HN$_{CoNiC}$ (0.711). Epithelial cells remain more consistent, but HN$_{Large}$ achieves the highest balanced accuracy with 0.796, HN$_{CoNiC}$ the lowest with 0.789. For segmentation quality, HN$_{CoNiC}$ achieves the lowest Hausdorff distances across all cell types except eosinophils (Figure 2 A). Investigating EosEval, we find an increase in performance across all model sizes compared to GlaS. HN$_{Large}$ has 0.553 F1 score on GlaS, but a per region average F1 score of 0.668 on EosEval. On MitEval, HN$_{Tiny}$ performs best with 0.553 F1 compared to 0.521 for HN$_{Large}$ and 0.517 for HN$_{Base}$. Evaluating the effect of TTA, we observe an initial F1 score bump when using TTAs (Figure 2 C), but less of an increase with additional views. HN$_{Base}$ has an increase of 2.4% in F1 from zero to four

TTA views, but less from eight to sixteen (+0.0004) with standard deviation decreasing slightly (+-0.0039 to +-0.0036). Additionally, we find that the rare cell types, eosinophils, neutrophils, and mitoses gain the most from TTA (+2.16, +4.16, +4.87 % F1 Score with 4 TTA), whereas more common cell types only show increases of less than one percent or even a slight negative impact (Plasma cells: -0.35%).

### 3.2. PanNuke

We train HoVer-NeXt on PanNuke, optimize hyperparameters, and evaluate it with the PanNuke evaluation script (Gamper et al., 2020). $HN_{Tiny}$ with 16 TTA has a tissue average mPQ ($mPQ_{Tiss}$) of 0.477, achieves the highest PQ for inflammatory (0.418) and dead (0.154), and improves on HoVer-Net in the epithelium (+0.024) and connective (+0.027) class, yet only reaches 0.536PQ on neoplastic (Figure 3). CellViT sets the current state-of-

| Panoptic Quality | | | | | | | | *Model comparisons taken from the respective papers |
|---|---|---|---|---|---|---|---|---|
| | $bPQ_{Tiss}$ | $mPQ_{Tiss}$ | mPQ | Neo | Epi | Inf | Con | Dead |
| HoVer-Unet* | 0.629 | 0.448 | 0.372 | 0.524 | 0.478 | 0.401 | 0.379 | 0.076 |
| HoVer-Net* | 0.659 | 0.463 | 0.397 | 0.551 | 0.491 | 0.417 | 0.388 | 0.139 |
| $HN_{Tiny,TTA=4}$ | 0.651 | 0.473 | 0.404 | 0.531 | 0.510 | 0.416 | 0.411 | 0.151 |
| $HN_{Tiny,TTA=16}$ | 0.656 | 0.477 | 0.407 | 0.536 | 0.515 | **0.418** | 0.415 | **0.154** |
| CellViT-SAM-H* | **0.679** | **0.498** | **0.431** | **0.581** | **0.583** | 0.417 | **0.423** | 0.149 |

Figure 3: Average results over 3-fold cross-validation on PanNuke for different models

the-art with 0.498 $mPQ_{Tiss}$ with large improvements in neoplastic and epithelial panoptic quality (+0.03, +0.09) over HoVer-Net. However, Hörst et al. (2023) note that their model was only performing on par with HoVer-Net without pretraining. Moreover, we observe qualitatively, that the performance does not seem to translate to WSI (Supp. Figure 7). More thorough evaluation metrics for $HN_{Tiny}$ can be found in Supp. C.5.

### 3.3. Inference Time

We utilize five publicly available TCGA WSI (Supp. Table A.10) to evaluate inference timings for .24mpp and .5mpp for PanNuke and Lizard respectively (specs see Supp. A.9). $HN_{Large}$ trained on Lizard-mitosis at 0.5mpp takes 45s to run the smallest WSI and 7:26m

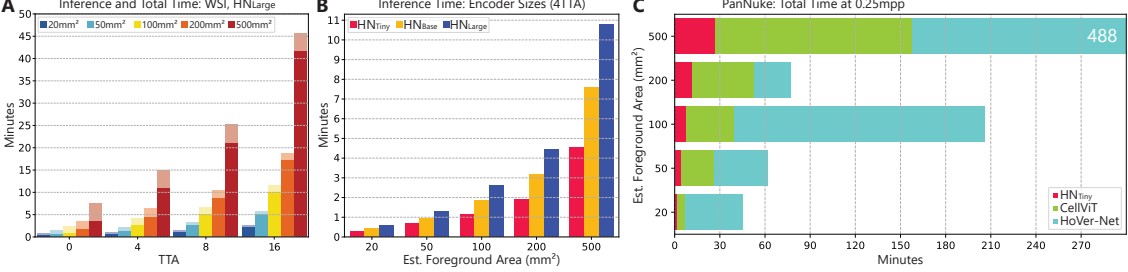

Figure 4: Inference timings across different images using $HN_{Large}$ (A), for different encoder sizes at 4TTA (B) and comparing HoVer-Net, CellViT and HoVer-NeXt (C)

for the largest WSI at 0 TTA. It takes ∼2× longer per 4 TTA views, with the largest WSI taking 14:52m at 4 TTA, and 25:12m at 8 TTA (Figure 4). Using the largest slide with 4 TTA as a reference, $HN_{Large}$ runs inference at 1.78s/mm². For PanNuke at 0.25mpp, $HN_{Tiny}$ takes 26:52m for the largest WSI, whereas CellViT and Hover-Net take 02:37:39h and 8:08:28h respectively. Based on the largest WSI, $HN_{Tiny}$ (at 4 TTA) processes WSI

at $3.22 s/mm^2$. The entire WSI test set takes 50:53m on $HN_{Tiny}$, 04:42:31h on CellViT, and 14:38:44h on HoVer-Net making our pipeline $5.6\times$ and $17.2\times$ faster. At the time of writing, HoVer-UNet had no WSI pipeline available, but with the reported $3\times$ speedup over HoVer-Net, it would take 4:52:54h.

## 4. Discussion

To make large-scale investigations into the cellular composition of CRCs feasible, we develop a pipeline for nuclei segmentation and classification. Our model retains the predictive performance of our original CoNiC challenge submission, improves upon the detection metrics, and successfully learns the additional mitosis class using the mitosis and Lizard-mitosis datasets. Differences in eosinophils and plasma cells between $HN_{CoNiC}$ and $HN_{Large}$ and the ablation results indicate that the loss function and sampling strategy have varied impact on rare cell types, however with no clear best configuration. We also find that particularly eosinophils and neutrophils are more sensitive to color changes, but reduce this problem with TTA. Also the removal of convex-hull-based post-processing likely leads to more segmentation outliers increasing the Hausdorff distance. One of the major improvements to further increase model accuracy on small datasets such as PanNuke are large-scale pretraining (Chen and Krishnan, 2022; Hörst et al., 2023) or semi-supervised learning approaches such as Rumberger et al. (2023). Also, other published implementations of ConvNeXt-based U-Net variants such as (Roy et al., 2023) could further improve results. Larger context sizes could also lead to more robust classification, in particular in the healthy vs. malignant case (Frei et al., 2023). As demonstrated by our adaptation of ST++ (Yang et al., 2021), automatic labeling of objects in histopathology by re-staining is a straightforward way of generating large labeled datasets and even single institute data provides sufficient variety for learning mitosis. Finally, HoVer-NeXt is $5\times$ faster than state-of-the-art and runs inference on TCGA COAD/READ (N=576) at 0.5mpp in 50 hours.

## 5. Limitations

Lizard contains wrongly annotated mitoses and not all pHH3-positive objects are visibly mitoses, thereby creating a noisy dataset. Moreover, perfect annotations on H&E are difficult, and cell types are only estimated by pathologists. Therefore, reported results will never entirely reflect the true model performance. Moreover, a more accurate approach for H&E mitosis annotations on Mit-Eval could have been chosen (Aubreville et al., 2023). Finally, we did not use the 3-fold evaluation split for Lizard to maximize available training data and compare with our own $HN_{CoNiC}$.

## 6. Conclusion

We publish HoVer-NeXt, a fast and efficient H&E-based nuclei segmentation and classification pipeline, allowing for investigations into cellular compositions, spatial relationships, and morphological parameters of nuclei directly on large cohorts. While much of current research focuses on spatial technologies such as multiplex-immunofluorescence, generating such data is still expensive. Here, HoVer-NeXt can be a used for hypothesis generation and for finding interesting WSI to be further investigated using spatial technologies. Our work facilitates the next generation of histopathology and provides an important building block towards a quantitative view of WSI in clinical routine.

## Acknowledgments

The results published here are in whole or part based upon data generated by the TCGA Research Network: https://www.cancer.gov/tcga. We'd like to thank Sophie Lechner for her support in annotating eosinophils, and Lucine Constance Christe and Philipp Zens for annotating Mitoses as additional observers. Calculations were performed on UBELIX (https://www.id.unibe.ch/hpc), the HPC cluster at the University of Bern.

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

## Supplementary Material

### A. Additional Details

#### A.1. Training

HoVer-NeXt is trained for 200000 steps with batch size 48 using AdamW optimizer with weight decay (0.0001). We use a cosine-annealing learning rate schedule from 1e-4 to 1e-8. For the encoder, we rely on imagenet pretrained ConvNeXt-v2 encoders from pytorch-image-models (Wightman, 2019). All encoders are trained with 50% dropout and decoder arms do not utilize dropout. As loss functions, the two arms for instance and semantic segmentation have separate specific losses. For the instance arm, the center point vector predictions are trained with MSELoss and the BCB map with cross entropy loss. The class prediction arm uses Focal Loss (gamma=2.0) and class and instance arm losses are summed and weighted using a weighting parameter (lambda = 0.02). Model selection is done via best validation metrics specific to the dataset instead of lowest validation loss. Based on recommendations from Tellez et al. (2019), we apply HED color augmentation, hue,saturation and brightness variation, random noise and Gaussian blurring. We also include random rotation, flipping, mirroring, zoom, scale, shear, translate and elastic transform. Post-processing during training for the validation step is done as explained in the inference pipeline section 2.3 except directly on tiles.

#### A.2. Resolving overlaps

A single worker stitches the ROIs to form the final output and resolves overlaps whenever there are nuclei in both the write space and the newly to be written ROI. Each side of the ROI is checked within 512px overlap regions to resolve duplicate instances or half-instances. All instances within the outermost quarter of the already written region will be kept as is and new instances in that area are discarded. Also if part of an old nucleus exists in the second quarter, it is kept as well and any information from the new ROI is discarded. Any other already written nuclei will be deleted and replaced by the new predictions from the second quarter onwards. Instance ID's are updated based on the largest previously written instance, but the instance numbering may not be contiguous. As the tiles are from the same inference process, there will be no differences in class assignments and this method will only be problematic if an instance is larger than the overlap region, but this is not the case in the investigated datasets and domains.

#### A.3. Foreground Background Estimation

We estimate the foreground of whole slide images on the thumbnail of the WSI which is available via OpenSlide in all common WSI formats. The thumbnail size is dependent on the WSI and Scanner and ranges from 1/75 to 1/160 of the full resolution image and the final mask information is rescaled to the required image size depending on the retrieved tile magnification. The thumbnail is first converted to gray scale using the conversion matrix of OpenCV and subsequently blurred with a $5 \times 5$ averaging kernel to avoid high frequencies and noise. We set an intensity threshold of 240 and keep all pixels below that threshold forming one ore multiple foreground regions. Then we filter the foreground regions by removing objects that are smaller than 0.01% of the image and finally expand all kept regions with

a dilation step using a circular kernel. The size of the kernel is again determined by the image dimensions as we use the 0.01% of the size of the longest dimension as diameter. The filtering step ensures that we do not keep small fragments and small slide artifacts as relevant foreground and the dilation step avoids cut corners where some lighter tissue would be missing due to the blurring step. These threshold steps were chosen qualitatively by considering WSI from multiple cohorts and verifying that estimated foreground area was within reasonable bounds.

### A.4. On the potential negative effects of test-time augmentations

Applied augmentation methods were selected during the original challenge submission with all transformation parameters being chosen such that the transformed images still appear as though they could be crops form an H&E image. However, some of these transformations remove information or make it more challenging for the model to make a correct prediction such as adding Gaussian noise or blurring the image. Therefore these transformations are removed from the set of augmentation methods during inference, but during training it is useful for the model to also learn to produce acceptable results even if the image is blurry. Including during training are also elastic deformation, rotation in a range from $0° - 45°$, shearing, as well as zooming, all of which utilize an interpolation method to transform the image, thereby changing image information. The same then applies for the model outputs, which need to be inversely transformed, where then any differences introduced by the interpolation will lead to less exact nucleus boundary predictions. Additionally, rotating by $45°$ or shifting the image removes pixels completely which also leads unnecessarily worse performance. Figure 5 illustrates these concepts.

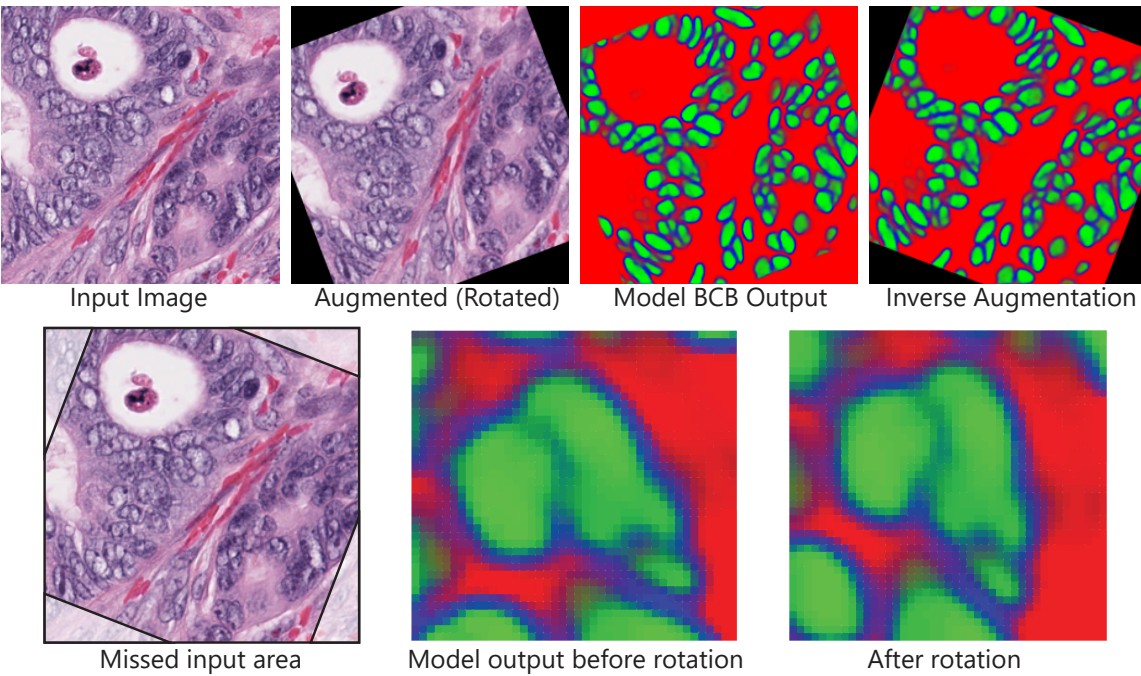

Figure 5: Example of a potentially problematic augmentation method: The input image is rotated by less than 90°which means that an interpolation method needs to be used. The model then receives the transformed image, provides an output but naturally cannot provide an output for the now invisible area. When rotating the image back to its original orientation, a large area of the original input is actually missed. Zooming into a detail of the rotation of the model output, we can quickly observe differences where the interpolation method softens some edges and reduces differences between neighboring pixel values. While this does not necessarily have to have a negative impact on the final result, differences of a single pixel can already change the hausdorff distance for this nucleus significantly.

## A.5. LIZARD AND PANNUKE DESCRIPTION

Both datasets are available at the TIA-Warwick website

**Lizard** The Lizard dataset is an H&E based nuclei segmentation and classification dataset for CRC and normal colon tissue with six classes: neutrophils, epithelial cells, lymphocytes, plasma cells, eosinophils, and connective tissue cells (Graham et al., 2021a). Raw H&E images are available at 0.5mpp both as pre-cropped (with overlap) $256 \times 256$ tiles as well as full ROIs. It combines multiple datasets from several institutes and has 495,179 total annotated nuclei but is highly imbalanced with neutrophils and eosinophils only accounting for 0.89% and 0.68% of all instances. Additionally, ∼84% of the dataset is background. The dataset combines multiple datasets from different institutes, one of which is the GlaS subset which we are using as an external test set.

**PanNuke** PanNuke is another H&E nuclei panoptic segmentation dataset but with a wider focus on samples from many different cancers (Gamper et al., 2019, 2020). Here the classes are neoplastic cells, inflammatory cells, connective tissue cells, dead cells, and non-neoplastic epithelial cells, again with considerable class imbalance as well as tissue type imbalance. PanNuke is only available as $256 \times 256$px crops and only at $\sim$0.25mpp.

### A.6. REGISTRATION AND GROUND TRUTH PREPARATION FOR THE pHH3 MITOSIS DATASET

All whole slide images are converted to TIFFs. Then, each H&E - pHH3 pair is registered in its entirety by manually specifying an anchor point in an exactly matching tissue area (e.g. by selecting a nucleus that was clearly observable in both images) to remove any offset differences in the images. Afterwards, we use SimpleElastix to estimate rigid and non-rigid registration transforms using greyscale versions of the images down-sampled to 0.5mpp and apply these transforms to the whole slide images at full resolution. All registrations were performed on a machine with 64Cores and 512GB memory. Thresholds for ground truth masks from the DAB channel are set individually per ROI to account for intra- and inter-WSI differences. ROIs were deliberately selected to include a large area of potential mitoses and areas problematic for the pHH3 stain such as necrotic areas or clear stain artifacts were avoided. All registered images, ROIs and generated masks were verified qualitatively to ensure that accurate segmentation masks were generated.

### A.7. SELF-TRAINING FOR MITOSIS

First, we train five models on just the lizard dataset with 5 cross validation folds and run ensemble inference on the the new mitosis crops. Then, a new model is trained on the combined dataset and checkpoints are saved every 50000 steps. Based on relative per sample changes in mean panoptic quality from the first checkpoint to the best (considering validation metrics) checkpoint, samples are split into easy and hard samples. Easy samples would be the ones with less than median change in panoptic quality and hard samples those with more. The same model is then re-trained from scratch only on the easy samples from the mitosis dataset and used to predict the hard samples creating the final mitosis dataset. Mitosis ground truth from the restain are always the only "nuclei" of the mitosis class and mitosis predictions are re-classified to the second most likely class. A model trained on only the mitosis dataset is then used to predict mitoses on Lizard where new mitosis annotations are only added if there is no other label on any of the pixels yet.

### A.8. ADDITIONAL VALIDATION: MitEval AND EosEval

**Mitosis Evaluation** For this dataset, we specify 13 ROIS from nine randomly selected CRC H&E resection WSI to ensure that each mitosis can actually be observed on the H&E which is not guaranteed when using automatic label generation from pHH3-based restains. pHH3 is also positive for cells in G2 and some other objects also sometimes pick up the antibody. Nine ROIs are on five slides from an internal cohort (0.12mpp) and four ROIs are on four publicly available TCGA Slides (0.25mpp). Annotations were done as small ellipses around the mitotic figures by three pathologists. The final dataset matches annotations by a maximum distance of $6\mu m$ and similar to Aubreville et al. (2023), annotations with

at least one matching additional annotation are added to the dataset, however we do not include an additional review step. The three observers have an ICC3 of 0.860 [0.69,0.95]. Images at ∼0.5mpp are provided both pre-tiled, and as complete ROIs in npy format.

**Eosinophil Evaluation**   Eosinophils are a comparatively easy to spot subset of immune cells discernible on H&E, yet in the lizard dataset, and during the CoNiC challenge, none of the models perform well in detecting them. Therefore, we created an additional eosinophil point annotation dataset with 11 ROIs of varied sizes from 8 Patients to further evaluate eosinophil detection performance, in particular also across different stain variations. ROI raw H&E images (at ∼0.5mpp) and annotations are provided as individual npy files.

### A.9. SPECIFICATIONS FOR INFERENCE TIME COMPARISON

All models run on a HPC Node with 2xIntel(R) Xeon(R) Gold 6248 CPU @ 2.50GHz, (20 cores each), 1600GB RAM with one NVIDIA A40 48GB GDDR6. We set the following parameters for CellViT, HoVer-Net and HoVer-NeXt, unspecified parameters are left to default. In our HPC environment, we need to set OMP_NUM_THREADS to 16 (matching the number of workers) for the PyTorch HoVer-Net pipeline (github.com/vqdang/hover_net) as we otherwise could not achieve competitive speeds. For CellViT, all TCGA-AA* images had to be ran using the 20× parameter and the others with the 40× parameter set. While the images indeed have different magnifications stored, their resolution is the same (∼0.25mpp). Moreover, the CellViT Pipeline requires pre-processing WSI which we included into the total processing time as the other two pipelines do this on the fly. All experiments are run on only 16 cores. For future evaluation, we also specify the commit and repository used for the comparison:
HoVer-Net (67e2ce5e3f1a64a2ece77ad1c24233653a9e0901)
CellViT (4bc42811c9841805ef0984b3ec0daf159312323a)
HoVer-NeXt (c5bf99fdc2d8bd5129d780c5f19ee83a4babb0d4).

| HoVer-Net | CellViT | HoVer-NeXt |
|---|---|---|
| –batch_size=64 | –batch_size 30 | –batch_size=64 |
| –model_mode=fast | –gpu 0 | –tta 4 |
| –nr_inference_workers=16 | –magnification 20/40 | –inf_workers 16 |
| –nr_post_proc_workers=16 | –geojson | –pp_workers 16 |
| | –enforce_amp | –overlap 0.9375 |

## A.10. Inference time comparison slides (TCGA)

Selected sample Images for inference speed comparisons. Images were selected based on the identified foreground area and limited artefacts on the slide. They are supposed to represent different tissue sizes from $20\text{mm}^2$ to $500\text{mm}^2$ thereby being examples of realistic applications from biopsy to resection blocks. All Images are from the TCGA-COAD/READ cohorts and can be obtained from https://portal.gdc.cancer.gov/. Foreground estimates are computed using our internal foreground estimation pipeline, the internal FGBG estimation of HoVer-Net computes smaller foreground areas, in particular for the larger wsi.

| Case ID | Slide ID | mpp | est. fg size | HoVer-Net est. fg. |
|---------|----------|-----|--------------|--------------------|
| TCGA-AA-3977 | DX1 | 0.2325 | $20.16\text{mm}^2$ | $19.85\text{mm}^2$ |
| TCGA-AA-3688 | DX1 | 0.2325 | $49.84\text{mm}^2$ | $46.74\text{mm}^2$ |
| TCGA-AA-A010 | DX1 | 0.2325 | $101.09\text{mm}^2$ | $99.97\text{mm}^2$ |
| TCGA-CK-4951 | DX1 | 0.2520 | $202.90\text{mm}^2$ | $137.46\text{mm}^2$ |
| TCGA-5M-AAT5 | DX1 | 0.2525 | $501.00\text{mm}^2$ | $421.18\text{mm}^2$ |

## B. Additional Figures

### B.1. TILE ARTEFACTS ON HN-CoNIC COMPARED TO HN-LARGE

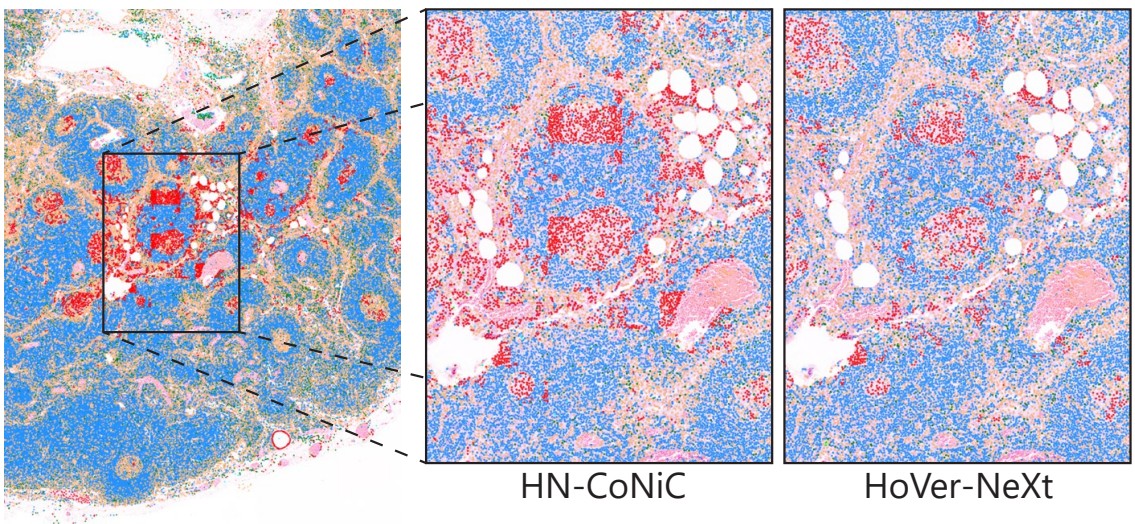

HN-CoNiC        HoVer-NeXt

Figure 6: In the images, we can see the same region from a cancer-free lymph-node with predictions from $HN_{CoNIC}$ and $HN_{Large,TTA=4}$. We note that either way, the epithelial predictions are wrong, however we highlight the reduction in tile based processing artefacts and overall reduction in false positive epithelium predictions. These tile based artefacts occur mostly if the tile normalization metrics transform the tile in way unseen during training. Most of the tiles in the training set do not really contain background, tiles with lymphocytes rarely show a germinal center and other strong color expressions such as ink or blood are also absent. Therefore, we recommend a constant normalization for 8bit RGB images both during training as well as during inference

### B.2. Qualitative comparison of HoVer-Net, CellViT, and HoVer-NeXt

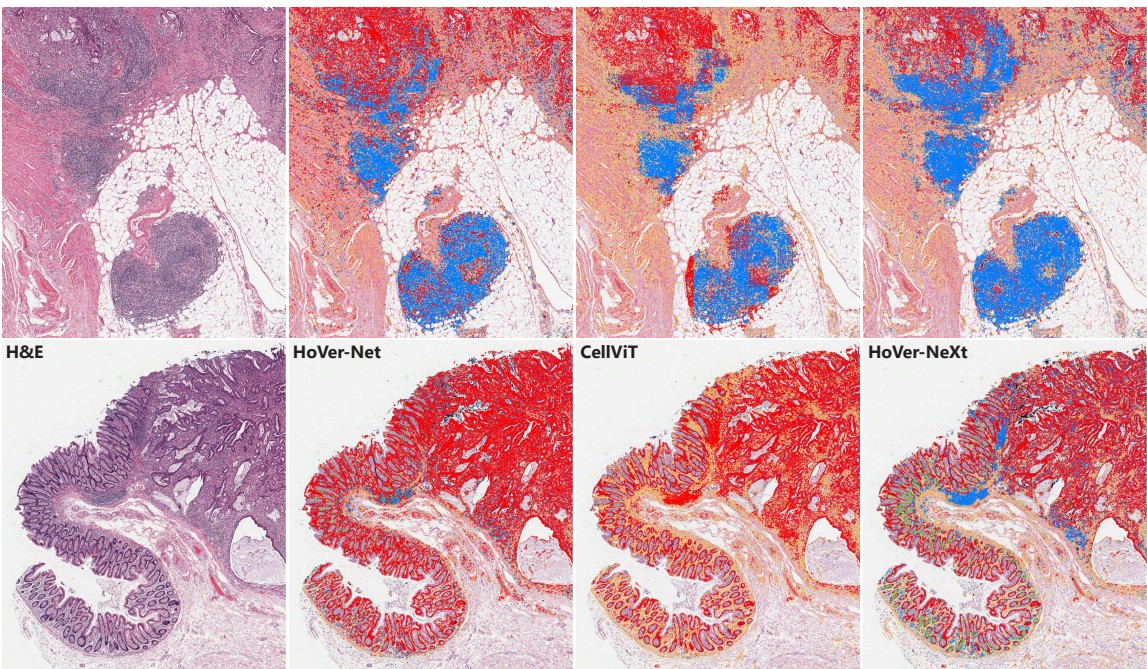

Figure 7: Qualitative comparison of HoVer-Net, CellViT, and HoVer-NeXt. Here we only consider detections, an not segmentations. For CellViT observe the same square patterns as we saw with HN-CoNiC (App. Figure 6 and an tertiary lymphoid structures completely predicted as neoplastic epithelium. HoVer-Net filters the entire normal submucosa for processing and overpredicts neoplastic cells in general. HoVer-NeXt predicts some of the normal (perhaps hyperplastic) mucosa as normal epithelium, but also falsely classifies a lot of it as neoplastic. It is the only model that classifies the lymphoid aggregates mostly correctly, yet also misclassifies some vessels and histiocytes as neoplastic epithelium.

## C. Additional Tables

### C.1. AUGMENTATION PARAMETERS

Parameters for applied augmentation methods during training and for test-time augmentations. Color augmentations are performed in part with custom functions written in pytorch and are defined by a single scaling factor that adjusts all parameters, however we report the individual scaled values. The HED augmentation method is adapted from Tellez et al. (2019). Spatial augmentations rely on a custom augmentation module written entirely using pytorch functions. All augmentation methods run on GPU.

| Method | Train Parameters | p(Train) | Test Parameters | p(Test) |
|---|---|---|---|---|
| Color Aug. | | | | |
| Color Jitter | B,C,S,H=[0.32,0.32,0.2,0.08] | 0.3 | - | - |
| HED Aug. | $\sigma$=0.03, bias=0.05 | 0.75 | $\sigma$=0.03, bias=0.05 | 1.0 |
| Gaussian Noise | $\sigma$=0.05 | 0.3 | - | - |
| Gaussian Blur | size=15, $\sigma$=(0.1,2.0) | 2.0 | - | - |
| Spatial Aug. | | | | |
| Mirror | H(p=0.5),V(p=0.5) | 0.5 | H(p=0.5),V(p=0.5) | 0.5 |
| Translate | Max pct.=0.05 | 0.2 | - | - |
| Scale | Min=0.8,Max=1.2 | 0.2 | - | - |
| Zoom | Min=0.5,Max=1.5 | 0.2 | - | - |
| Rotate | Max deg.=179° | 0.75 | Only 90° | 0.75 |
| Shear | Max pct.=0.1 | 0.2 | - | - |
| Elastic | $\alpha = (120,120)$, $\sigma$=8 | 0.5 | - | - |

## C.2. Ablation Study: Sampling vs. Weighting

| LW | DS | Metric | | | Neu | Epi | Lym | Pla | Eos | Con |
|---|---|---|---|---|---|---|---|---|---|---|
| | | Bal. Acc. | | mAcc. | Neu | Epi | Lym | Pla | Eos | Con |
| ✓ | ✓ | | | 0.762 | **0.647** | **0.798** | 0.857 | **0.726** | 0.737 | 0.803 |
| | ✓ | | | 0.759 | 0.627 | 0.796 | **0.858** | 0.693 | **0.785** | 0.794 |
| ✓ | | | | **0.766** | 0.641 | 0.789 | 0.855 | 0.716 | 0.772 | **0.816** |
| | | | | 0.755 | 0.617 | 0.789 | 0.847 | 0.697 | 0.776 | 0.803 |
| | | HD | | | Neu | Epi | Lym | Pla | Eos | Con |
| ✓ | ✓ | | | | **2.109** | 2.856 | 1.168 | 1.205 | 2.239 | 2.032 |
| | ✓ | | | | 2.250 | 2.843 | 1.165 | **1.196** | **2.149** | **2.003** |
| ✓ | | | | | 2.283 | **2.821** | **1.161** | 1.211 | 2.255 | 2.027 |
| | | | | | 2.543 | 2.980 | 1.327 | 1.416 | 2.477 | 2.137 |
| | | F1 | bF1 | mF1 | Neu | Epi | Lym | Pla | Eos | Con |
| ✓ | ✓ | | 0.844 | **0.607** | 0.293 | 0.829 | 0.765 | 0.493 | 0.544 | 0.718 |
| | ✓ | | 0.841 | 0.606 | **0.313** | 0.826 | 0.766 | 0.471 | **0.553** | 0.708 |
| ✓ | | | **0.846** | 0.605 | 0.254 | **0.830** | **0.767** | **0.501** | 0.551 | **0.729** |
| | | | 0.836 | 0.571 | 0.196 | 0.820 | 0.749 | 0.442 | 0.506 | 0.713 |
| | | PQ | bPQ | mPQ | Neu | Epi | Lym | Pla | Eos | Con |
| ✓ | ✓ | | 0.538 | 0.453 | 0.197 | 0.608 | 0.638 | 0.404 | 0.353 | 0.517 |
| | ✓ | | **0.546** | **0.454** | **0.206** | 0.606 | **0.644** | 0.386 | **0.369** | 0.516 |
| ✓ | | | 0.543 | 0.452 | 0.161 | **0.611** | 0.642 | **0.411** | 0.359 | **0.526** |
| | | | 0.518 | 0.414 | 0.119 | 0.593 | 0.607 | 0.346 | 0.312 | 0.506 |
| | | Binary Px. | bF1 | bMCC | | | | | | |
| ✓ | ✓ | | **0.821** | 0.784 | | | | | | |
| | ✓ | | 0.819 | **0.786** | | | | | | |
| ✓ | | | 0.821 | 0.783 | | | | | | |
| | | | 0.814 | 0.776 | | | | | | |

Table 1: Ablation Study: Class based loss weighting (LW) (Using the same focal loss with esimated class weights from Rumberger et al. (2022)) and class distribution based data sampling (DS) in comparison. $HN_{Large}$ with 16TTA, fixed seed, 10 run average. If two values in this table are the same, the omitted decimals are used for deciding which is best.

## C.3. MitEval full metrics

To evaluate this dataset, we use the same distance based matching using the centroid of the annotated ellipse.

Mitosis detection

|  | Precision | Recall | F1 |
|---|---|---|---|
| HN$_{Large}$ | 0.564125 | 0.680488 | 0.616874 |
| HN$_{Base}$ | 0.527298 | 0.670855 | 0.590480 |
| HN$_{Tiny}$ | 0.545022 | 0.720167 | 0.620478 |

## C.4. EosEval full metrics

As the eosinophil test-set only consists of point annotations, we cannot compare any segmentation metrics and only report detection measures. Results are averaged (std) over 7 patients with 11 different ROIs in total

Eosinophil detection:

|  | Precision | Recall | F1 |
|---|---|---|---|
| Large | 0.699 +- 0.120 | 0.695 +- 0.061 | 0.688 +- 0.055 |
| Base | 0.696 +- 0.128 | 0.687 +- 0.073 | 0.681 +- 0.063 |
| Tiny | 0.641 +- 0.155 | 0.700 +- 0.072 | 0.654 +- 0.083 |

## C.5. PanNuke: Additional Results

All results are averaged over the three official folds without center-cropping. In practice, results are most likely better.

| Binary px. metrics |  |  |
|---|---|---|
|  | F1 | MCC |
| HN$_{Tiny,4TTA}$ | 0.802 | 0.810 |
| HN$_{Tiny,16TTA}$ | 0.803 | 0.811 |

| F1 Score |  |  |  |  |  |  |  |
|---|---|---|---|---|---|---|---|
|  | bF1 | mF1 | F1$_{Neo}$ | F1$_{Epi}$ | F1$_{Inf}$ | F1$_{Con}$ | F1$_{Dead}$ |
| HN$_{Tiny,4TTA}$ | 0.822 | 0.649 | 0.715 | 0.723 | 0.679 | 0.641 | 0.486 |
| HN$_{Tiny,16TTA}$ | 0.826 | 0.653 | 0.720 | 0.728 | 0.681 | 0.646 | 0.492 |

| Hausdorff Distance | Neo | Epi | Inf | Con | Dead |
|---|---|---|---|---|---|
| $HN_{Tiny,4TTA}$ | 5.683 | 5.958 | 3.729 | 5.132 | 3.570 |
| $HN_{Tiny,16TTA}$ | 5.622 | 5.918 | 3.717 | 5.090 | 3.551 |

| Balanced Accuracy | mAcc | Neo | Epi | Inf | Con | Dead |
|---|---|---|---|---|---|---|
| $HN_{Tiny,4TTA}$ | 0.779 | 0.760 | 0.852 | 0.813 | 0.748 | 0.723 |
| $HN_{Tiny,16TTA}$ | 0.782 | 0.764 | 0.854 | 0.814 | 0.751 | 0.725 |

| Tissue Average | bPQ | | mPQ | |
|---|---|---|---|---|
| | $HN_{Tiny,4TTA}$ | $HN_{Tiny,16TTA}$ | $HN_{Tiny,4TTA}$ | $HN_{Tiny,16TTA}$ |
| Adrenal gland | 0.702089 | 0.703924 | 0.49439 | 0.494484 |
| Bile-duct | 0.665283 | 0.667678 | 0.465224 | 0.467801 |
| Bladder | 0.693248 | 0.696374 | 0.575255 | 0.578479 |
| Breast | 0.640457 | 0.643159 | 0.493503 | 0.49549 |
| Cervix | 0.665073 | 0.666972 | 0.474109 | 0.47509 |
| Colon | 0.566692 | 0.570241 | 0.425545 | 0.428342 |
| Esophagus | 0.644828 | 0.64745 | 0.524058 | 0.52689 |
| Head&Neck | 0.641031 | 0.643037 | 0.481729 | 0.484619 |
| Kidney | 0.6809 | 0.683341 | 0.512658 | 0.51673 |
| Liver | 0.715341 | 0.716678 | 0.501918 | 0.504076 |
| Lung | 0.630222 | 0.634101 | 0.425785 | 0.428984 |
| Ovarian | 0.608334 | 0.611863 | 0.483388 | 0.485762 |
| Pancreatic | 0.655729 | 0.657374 | 0.45788 | 0.460296 |
| Prostate | 0.62628 | 0.628754 | 0.480863 | 0.480669 |
| Skin | 0.620624 | 0.622956 | 0.410657 | 0.414369 |
| Stomach | 0.694477 | 0.696453 | 0.458618 | 0.461314 |
| Testis | 0.678664 | 0.679845 | 0.497335 | 0.49749 |
| Thyroid | 0.675996 | 0.67747 | 0.420411 | 0.422295 |
| Uterus | 0.617216 | 0.618833 | 0.44565 | 0.446299 |

