# OpenReview forum: "HoVer-NeXt: A Fast Nuclei Segmentation and Classification Pipeline for Next Generation Histopathology"
_MIDL.io/2024/Conference — MIDL 2024 Oral_

### Official Review · Reviewer_wR9R · 2024-02-21

**Confidence:** 4
**Preliminary Rating:** 4
**Recommendation:** Poster
**Final Rating:** 5

**Summary:**

The presented work describes the development of a model for nuclei segmentation and classification for the quantitative assessment of tissue composition in histopathology. The model is described as an updated version to a previous challenge submission that reported competitive performance. In comparison to the already published challenge submission, the authors updated the model with some extensions and simplifications, provide new training and validation data for the scientific community, validate their updated model on the PanNuke benchmark dataset for comparison and provide an efficient WSI inference pipeline, which they evaluate against other relevant methods from the literature.

**Strengths:**

- The authors provide all code for the training and inference pipeline of their model which can be of great value to the scientific community.
- Additionally, they also provide all additional data that was used to evaluate their models, including the model weights which promotes transparency and reproducibility.
- Their developed WSI inference pipeline demonstrates great time savings compared to already existing methods while showing competitive performance.
- The authors also provide extensive evaluation metrics and implementation details to validate their experiments.
- Overall, the manuscript is well written.

**Weaknesses:**

While the authors acknowledge that the newly added mitoses class is a noisy dataset in the limitation section, the validation of the reliability of these added labels could have been more elaborate.
- First, quantifying mitotic figures is a very challenging task even for experienced pathologists. Therefore, evaluating the model on a hold-out test set which was annotated by a single pathologist can create a very biased result, as inter-rater agreement between pathologists can be quite low. This risk of misidentifying mitotic figures is even elevated at a resolution of 0.5mpp as small differences in the morphology are more difficult to distinguish.
- Second, as mentioned by the authors, the creation of labels from PHH3 poses some additional challenges such as selecting a proper threshold for DAB channel, but also properly registering the PHH3 annotations to the HE image, which methodology was not mentioned by the authors in the manuscript. While the self-training strategy describes a sophisticated way of generating labels in this way, the resulting labels will still contain mislabeled mitotic figures and mitotic figures not visible in HE.
- Additionally, mitotic figures were only added to the Lizard dataset when no other label was on any of the pixels yet, which is questionable, as it is likely that some of the mitotic figures were previously misassigned to one of the original classes, and therefore additional label inconsistencies are introduced by the new mitosis class. The new mitosis class in the Lizard-Mitosis dataset were not verified by any pathologist and its reliability is therefore questionable.
- Finally, the authors claim that the single institute data for mitosis was sufficient to learn mitosis, however the experiments do not support this claim, as the training data included only ROIs from 11 samples and there was, e.g., no evaluation of the mitosis class on data from a different institute. Therefore, the robustness of this method is questionable and this should be acknowledged.

The authors claim to have achieved a new SOTA performance with the PanNuke benchmark dataset. However, the CellViT model has better validation metrics. The authors claimed that these results were only achieved by the CellViT model when they used pre-trained models. The use of pre-trained models for the encoder should not invalidate the performance. Especially for transformers, pre-training strategies on large datasets are of high importance and the use of publicly available models should be encouraged. Therefore, the CellViT model should not be denied SOTA performance.

**Detailed Comments:**

There is a typo in the header of Figure 2D. The title says Mitosis Dataset Evaluation twice. One of them should be Eosinophil Dataset Evaluation.

**Justification Of Final Rating:**

The authors put significant effort into the rebuttal and the revision of the manuscript, which significantly benefitted the manuscript and removed all previous doubts. This has become a really great paper, and I am certain this needs to be presented at MIDL.

**Justification Of The Preliminary Rating:**

The authors have provided an extensive evaluation of the updated model from their previous challenge submission and improved the efficiency of the inference for WSI. However, the addition of the new mitosis class for the Lizard dataset is not validated rigorously and introduces label noise.

**Questions To Address In The Rebuttal:**

- How reliable are the mitosis labels in the Lizard-Mitosis dataset?
- How often was the mitosis label predicted for a cell that was previously assigned to another class?
- Is it therefore sensible to only include the new mitosis label, if no labels were assigned to the pixels yet.
- How reliable are the mitosis annotations created by the pathologist?
- Is it possible to create a multi-rater consensus for the hold-out test set or use another dataset which provides labels as multi expert consensus?

---

> ### Author Response · Authors · 2024-03-17
>
> We would like to thank the reviewer for the time invested in reviewing our manuscript. We feel the paper is now significantly improved and we have done our best to address them in changes in the manuscript as well as the following responses.
>
> Regarding the missing information on the image registration
>
> To address these questions, we have added an additional section in the supplementary material (A.6.) explaining the process used for image registration and DAB label extraction. Image registration is done with the entire WSI using SimpleElastix and a single manually defined anchor point to pre-align images to ensure a correct registration. Given the size of the images, registration is done at 0.5mpp and then applied to the full resolution images.
>
> Regarding CellViT’s SOTA:
>
> We agree with the arguments you put forward in that CellViT should not be denied SOTA and have therefore updated the tables and re-worded our manuscript to reflect these changes.
>
> “How reliable are the mitosis labels in the Lizard-Mitosis dataset?”
> And
>  “How often was the mitosis label predicted for a cell that was previously assigned to another class?”
>
> The reason we included an additional mitosis test set is to avoid the most likely impossible task of quality controlling each mitosis label. Lizard is only available in form of small ROIs, there is no context which can be used to generate a general impression of how mitotic figures look like in these samples and all ROIs are only available at 0.5mpp.
>
> There are 331 newly added mitoses in the lizard dataset. The predicted labels contained 407 mitoses, which means 76 objects were predicted as mitosis but already had a different label in lizard. 55 were labelled as lymphocyte, 11 as epithelium, and 5 each as neutrophil, plasma cell and connective tissue cell. That means 76 objects might be mitoses, but a pathologist decided differently, with 331 objects remaining as “potential” mitoses. Correcting and verifying this manually suffers from the exact issues the reviewer is describing in that these images are only available at 0.5mpp with varying stain and scanning quality.
>
> Our method using self-training was intended to adjust for the noisy mitosis labels by using phh3 ground truth in another dataset, which in turn has noisy (i.e. not verified) labels for the other classes. During training a batch is drawn randomly from one or the other dataset.
> In ST++ paper, the authors suggest that their method even improves the model’s results on the test-set which is not necessarily something we could verify, but we note that this in turn means that the noise introduced by this self-training process does not have a negative effect on the final results. Of course, a nuclei segmentation and classification dataset with ground truth IHC labels for all classes would be more optimal, but we show that we can add this rare class using a separate dataset.
>
>
> "Is it therefore sensible to only include the new mitosis label, if no labels were assigned to the pixels yet."
>
> If we overwrote an existing label, we would have to make sure the model is correct, however the authors of the lizard dataset report that annotations were verified by pathologists, which we would consider a more accurate annotator than our own model. As already mentioned, the images are only available at low resolution and potentially more compressed than the original slides which the original annotators could have used to make a better decision. On a more fundamental note, we believe that creating a large scale dataset for H&E object annotations based on nuclei/object specific antibodies with restains for all classes might be a necessary next step to further improve H&E based nuclei classification and to better validate the results.
>
>
> Regarding
>     “How reliable are the mitosis annotations created by the pathologist? “
> And
>    “ Is it possible to create a multi-rater consensus for the hold-out test set or use another dataset which provides labels as multi expert consensus?”
>
> We have added annotations by two more pathologists, updated the supplementary material accordingly and updated the mitosis detection results by taking majority vote labels from the three pathologists. Their ICC3 is 0.860 [0.69,0.95]. We have added to our limitations section, that a more optimal way of annotating could have chosen as e.g. shown by Aubreville et al. (2023), but given the limit time scope of this rebuttal period, we at least have a majority consensus on all labels now.
>
> Additionally, we moved the MitEval dataset to the same ROI-WSI-aggregation evaluation as the eosinophil dataset, such that metrics from each different WSI contribute equally to the final metric and we can show between slide variability. Plots and descriptions have been updated accordingly.
>
> Lastly, we have fixed the wrong label for the mitosis dataset evaluation.

---

> > ### Comment · Reviewer_wR9R · 2024-03-21
> > **Response to the authors**
> >
> > Thank you for your thoughtful response to my review. I appreciate the time and effort you invested in addressing my concerns.
> >
> > 1. Missing Image Registration
> >
> > The addition of a new section (A.6) in the supplementary material explaining image registration and DAB label extraction is a valuable addition. This context will undoubtedly benefit readers in understanding your methodology.
> >
> > 2. CellViTs SOTA
> >
> > I acknowledge your point that the strengths and potential of your work will be evident to attentive readers, despite revisions made.
> >
> > 3. Quality of Mitosis Labels
> >
> > I appreciate the significant effort taken to improve the validity of the test set results by incorporating annotations from two additional pathologists and achieving a strong inter-rater agreement (ICC3  = 0.860). While acknowledging the limitations of using unverified labels in the lizard dataset, I recognize your emphasis on these limitations within the manuscript. This transparency allows readers to interpret the findings appropriately.
> >
> > Overall, the revisions you implemented effectively address the concerns I raised. Thank you for strengthening your manuscript.

---

### Official Review · Reviewer_KHyf · 2024-03-04

**Confidence:** 5
**Preliminary Rating:** 3
**Final Rating:** 5

**Summary:**

The authors introduce HoVer-NeXt, an updated version of their previous model, designed for scalable segmentation and classification of large whole slide images (WSI). This new model, along with its source code, has been made publicly available on GitHub. It includes the weights for ConvNextV2 trained on the Lizard-Mitosis dataset, with sizes detailed as Large (842.2MB), Base (391.6MB), and Tiny (144.4MB), in addition to a Tiny version trained on the PanNuke dataset. All the code and models are released under the GPL-3.0 license. Furthermore, the authors provide additional validation sets specifically for mitoses and eosinophils, enhancing the model's utility and accuracy in relevant biomedical analyses.

**Strengths:**

The paper is well-written, providing clear explanations of its methodologies and findings, which enhances its accessibility to a broad audience.

This model and its resources, including ConvNextV2 weights trained on Lizard-Mitosis and PanNuke datasets, are publicly available on GitHub under the GPL-3.0 license.

Additionally, they provide specialized validation sets for mitoses and eosinophils to improve accuracy in biomedical research. The extensive testing detailed in supplementary materials, alongside a robust training, validation, and testing methodology, underscores the model's reliability across various conditions.

**Weaknesses:**

The approach presented can be seen as a refinement of previously published works, positioning it more as a paper focused on optimization rather than introducing fundamentally new methodologies. It primarily enhances aspects related to cell segmentaion and classification, building upon the foundation laid by earlier research. This indicates that the contribution of the paper lies in its ability to optimize existing methods, rather than in the introduction of entirely new techniques or concepts in the field.

This work incorporates tunable hyper-parameters that are adjusted in a post hoc manner based on the validation sets. By including these adjustments as part of the end-to-end learning process, the overall performance and efficiency of the model are significantly enhanced.

The authors need to moderate their claims regarding the speed of their models, as the inference time comparison outlined in their 'A.6. Specifications for Inference Time Comparison' section does not constitute a strict apples-to-apples comparison. The discrepancies in the experimental setup (HoVer-NeXt is clearly favored), such as the unequal adjustment of hyper-parameters, inconsistent image processing parameters, underutilization of available computational resources, and reliance on default settings that may not be uniform across models, suggest that the observed differences in speed could stem, at least in part, from these technical inconsistencies rather than purely from the models' optimization. Therefore, a more cautious interpretation of the speed results is warranted until a more equitable comparison can be conducted.

**Detailed Comments:**

"... using a threshold on the HSV ... ": In section 2.3, regarding the Inference Pipeline, the document mentions utilizing a threshold on the HSV color space to process images. For clarity and to ensure the reproducibility of the results, it is essential to detail how this threshold was determined. Additionally, providing the specific threshold values used in the experiments would greatly aid in understanding and replicating the study's methods. This information is crucial for other researchers who may wish to apply the same approach or evaluate the model's performance under similar conditions. Such an analysis would provide a clearer understanding of how much added value each condition brings to the final model performance, offering a more nuanced view of the strategies employed to tackle label imbalance.

"...  the raw image at low resolution.": In the context of discussing the processing of the raw image at low resolution, it is important to specify the exact magnification level used—whether it is x40, x20, x10 or x5. This clarification is crucial for understanding the scale at which the images are analyzed and ensures the reproducibility of the results by providing precise details on the experimental setup.

"...  to avoid negative effects of augmentations.": The text mentions the need to avoid negative effects of augmentations, implying that certain image augmentation techniques can adversely affect the outcome of the analysis. It is important to elaborate on what these negative effects are. For instance, excessive or inappropriate augmentation methods might lead to overfitting, where the model learns the noise in the training data instead of the actual signal, reducing its ability to generalize to new, unseen data. Alternatively, certain augmentations could distort important features in the images, leading to misinterpretation by the model. Providing more details on these potential negative effects, including specific examples of augmentations that could be detrimental and the mechanisms by which they impact model performance, would greatly enhance the clarity and comprehensiveness of the discussion.

**Justification Of Final Rating:**

Thank you for the thorough revisions and additional details provided in response to the feedback, particularly on pipeline enhancements and model clarifications, which have addressed my initial concerns. The introduction of new studies and speed improvements significantly deepens the understanding and impact of your work.

Your efforts to differentiate your pipeline from existing models and the focus on efficiency for whole slide image analysis are commendable. The updates regarding the batch size and the implementation of automatic mixed precision during inference highlight the strength of your evaluation process.

I recommend a strategy to optimize your computational workflow to address the high memory demand, suggesting the use of dependency graphs and automated job arrays with OpenSlide's delayed reading feature. This would allow for more efficient memory management and flexibility in computational resource usage.

Given the comprehensive revisions and the significant contributions of your manuscript to the field, I have revised my decision to a strong accept. The advancements made in your work are commendable, and I look forward to its impact on future research and applications.

**Justification Of The Preliminary Rating:**

Given the concerns raised about the evaluation of HoVer-Net's efficiency, particularly in its comparison to other deep learning models for prediction tasks, my decision to give a borderline rating to this paper is grounded in several key issues. First, the efficiency evaluation of HoVer-Net, especially regarding inference times compared to its counterparts (HoVer-NeXt and CellViT), is not straightforward due to the significant differences in experimental setups. These differences include discrepancies in hyper-parameter adjustments, inconsistencies in image processing parameters, and the underutilization of computational resources, all of which potentially skew the comparison.

Moreover, the critique highlights the issue of relying on default settings, which vary widely among deep learning models, suggesting that observed differences in inference speed might not accurately reflect the models' efficiency but rather the conditions under which the experiments were conducted. This underlines the need for standardized conditions in performance comparisons to ensure any observed differences are due to the models' inherent capabilities and optimizations.

Furthermore, the choice of CellViT to operate with a much smaller batch size raises questions about the fairness and comparability of the performance evaluation. Batch size can significantly impact inference speed and efficiency, and using disparate batch sizes without clear justification introduces another variable that complicates direct comparisons.

Lastly, the suggestion to include architectures like Mask-RCNN, known for their optimization in handling heavy workloads, in the analysis indicates that a broader range of models should be considered for a fairer and more comprehensive performance evaluation. This inclusion would allow for a more balanced assessment of how different systems perform under demanding tasks, offering a clearer picture of their relative efficiencies.

In light of these concerns, my borderline decision is based on the need for a more rigorous, transparent, and standardized approach to evaluating and comparing the performance of deep learning models, ensuring that any conclusions drawn are robust, fair, and reflective of the models' true capabilities.

**Questions To Address In The Rebuttal:**

"... since data sampling is already sufficient to treat the label imbalance.": In section 2.2 on HoVer-NeXt, the text mentions that data sampling is adequate for addressing label imbalance. To substantiate this claim, it is recommended to present quantifiable metrics that clearly demonstrate the impact of each condition on the final results. An ablation study could be particularly insightful in this context. For example, the study could compare:

1. The combination of the model with class-based loss weighting and data sampling, illustrating how both techniques contribute together to mitigating label imbalance.
2. The model paired with class-based loss weighting alone, to assess the effect of adjusting the loss function to prioritize certain classes without altering the data distribution.
3. The model with data sampling only, to determine the effectiveness of this strategy in balancing the class distribution independently of loss function adjustments.

" ... To make HoVer-Net work at all, ...": The critique of the efficiency evaluation of HoVer-Net, particularly in its application to deep learning (DL) based predictions, warrants a more cautious interpretation. The comparison of inference times, as detailed in the section 'A.6. Specifications for Inference Time Comparison,' does not provide a straightforward comparison due to notable differences in the experimental setups (HoVer-NeXt is clearly favored). These differences encompass a range of issues, including unequal hyper-parameter adjustments, inconsistencies in image processing parameters, and the underutilization of computational resources. Such discrepancies undermine the credibility of the reported speed comparisons. Moreover, the reliance on default settings, which can vary significantly across different DL models, adds another layer of complexity to the comparison. This suggests that the differences in inference speed observed might not purely reflect the inherent efficiency of the models but could also be a result of the varied experimental conditions. Therefore, to accurately assess CellViT, HoVer-Net and HoVer-NeXt performance in DL-based predictions, it is essential to conduct comparisons under standardized conditions. This approach would ensure that any differences in speed observed are directly attributable to the models' optimization and capabilities, rather than external variables. Consequently, it becomes clear that the reported gains in speed may not solely derive from the deep learning model itself but from the entire processing pipeline in the HPC environment which is not fair.

Could you elucidate the rationale behind CellViT's implementation with a notably smaller batch size of 8, in contrast to the larger batch sizes of 64 employed by HoVer-Net and HoVer-NeXt?

For a more equitable comparison, it is suggested to incorporate architectures similar to Mask-RCNN into the analysis. Mask-RCNN is known for its optimization for handling heavy workloads effectively. Including such architectures would ensure that the comparison takes into account models that are well-suited for demanding tasks, leading to a fairer evaluation of performance across different systems.

---

> ### Author Response · Authors · 2024-03-17
> **Response Part 1**
>
> We thank the reviewer for the thorough review of our paper. The provided comments and feedback have helped to significantly improve the manuscript even further and are greatly appreciated. Please allow us to address each concern below.
> Importantly, we want to stress that indeed we present a pipeline here in this paper, and not the previously published model. While we have also made considerable changes to the model itself, the focus on many improvements lies on making our model more efficient and usable on whole slide images. Therefore, investigations and comparisons focus on the pipeline, and not the model.
>
> Regarding your specific raised issues and questions:
>
> “Additionally, providing the specific threshold values used in the experiments would greatly aid in understanding and replicating the study's methods.“
>
> Thank you for the comment. All threshold values for further postprocessing during model inference are specified in files that are included in the model weights zip file, as they are specifically optimized for each model. Hence, we believe an additional table summarizing these would be unnecessary.
>
> Regarding the questions on how foreground background estimation was performed:
>
> We have added a supplementary chapter on this topic to explain how FGBG estimation is performed in our pipeline, which can be found on page 14 of the revised manuscript (Supp. A.3.). We also corrected a mistake where we specified that the image is tresholded in HSV space, but rather it is done on the greyscale image using OpenCVs greyscale transform matrix. We note that the specific thresholds chosen are arbitrary in that, after trying a range of values, with the current values, we qualitatively could not observe any issues in missing foreground, and quantitatively could not observe outliers resulting from a bad foreground estimation on over 3000 slides from multiple different cohorts and institutions. Published works such as:
>
> Abbet, C., Studer, L., Zlobec, I., & Thiran, J. P. (2022, April). Toward automatic tumor-stroma ratio assessment for survival analysis in colorectal cancer. In Medical Imaging with Deep Learning.
>
>  use the same method. As this is not consistent with the method employed by HoVer-Net, we have also added a foreground area comparison into Supp. Table A.10.
>
> Regarding your points on the negative effects of augmentations
>
> Thank you for these comments. We have added another supplementary chapter (A.4) including a figure that further explains what we mean by reducing the augmentations during inference. Specifically, we want to highlight that some augmentations such as rotating the image by 45° removes information from the image which is necessary for a good segmentation. Naturally, during training, the ground-truth can be adapted accordingly to avoid this issue. Additionally, we have included a new table (Supp. C.1.). specifying all parameters for the augmentation methods, which was previously missing in the paper. The training augmentation strategy has been selected already during the CoNiC challenge submission and remains unchanged.

---

> > ### Author Response · Authors · 2024-03-17
> > **Response Part 2**
> >
> > Regarding the ablation study to compare loss weighting with sampling and the combination thereof:
> >
> > We added the requested ablation study as an additional chapter in the appendix C.2., which indeed does not provide sufficient evidence for differences in this dataset with this model between the methods for dealing with class imbalance. Only the model with no strategy to deal with class imbalance performs consistently worse than all other models. Therefore, we conclude from this limited information, that reducing overall complexity of the method by only using e.g. sampling or loss weighting, as suggested in our first draft, is still a reasonable approach with our claim that data sampling is sufficient still holding.
> >
> > Regarding your raised concerns that speed improvements might not solely be coming from improvements to the model or the HPC environment being tailored to our approach:
> >
> > We appreciate the reviewer’s concern. In this paper, we simplify and improve our previously published model, but mainly focus on creating a pipeline that can then be used for large scale investigations in existing H&E cohorts. In the context of nuclei segmentation and classification, the whole slide pipeline has the crucial step of resolving tile overlaps which we solve in a faster way than HoVer-Net (or CellViT). Moreover, tile extraction, memory management, and maximization of CPU and GPU utilization are all part of improvements of the pipeline that indeed are model independent. In contrast to other works, where algorithms are developed on pre-extracted tiles, we generate here an end-to-end pipeline for whole-slide inference.
> >
> > An apples-to-apples comparison in this setting is very difficult, as the reviewer correctly describes, as pipelines rely on different strategies for fast inference, with e.g. CellViT using larger tiles for inference and our method using less overlap than HoVer-Net. However, most end-users of such pipelines will be applying these models “as is” on WSI and we therefore aimed to create a comparison that will be valuable to end-users deciding on which pipeline to use for their project.
> >
> > Regarding your question on why we chose batch size 8 for CellViT:
> >
> > CellViT has a recommended input tile size of 1024x1024 compared to 256x256 of HoVer-Net and HoVer-NeXt. Batch size 8 is the default specification given by CellViT and we assume, similar to how we would specify a reasonable default implementation, that this batchsize results in a good performance. However, we did miss enabling automatic mixed precision during inference in the previous evaluation as this was indeed not enabled by default, as you correctly pointed out. As CellViT uses a SAM-based vision transformer encoder, the maximum possible batch size with mixed precision on a NVIDIA A40 GPU was 30, which is why the new evaluation now runs on this batch size. As this sped up the inference time, we have corrected our manuscript with updated timings and time comparisons. Moreover, we also noticed that we wrongly still included specifying omp_num_threads=1 and setting workers to 0, which we had already fixed prior to our latest results.
> >
> > Regarding your suggestion to compare with Mask-RCNN
> >
> > For a proper comparison with Mask-RCNN, we therefore would also need to create a full pipeline for Mask-RCNN which is not in the scope of this paper. Moreover, we note that past publications, such as HoVer-Net already include a comparison with Mask-RCNN in terms of model prediction quality and inference speed and conclude that HoVer-Net improves on it by a large margin:
> > “Regarding the processing time, the average time to process a 1000×1000 image tile over 10 runs using Mask-RCNN for segmentation and classification was 106.98 seconds. Meanwhile, HoVer-Net only took an average of 11.04 seconds to complete the same operation; approximately 9.7×  faster.”
> >
> > -	Graham, S., Vu, Q.D., Raza, S.E., Azam, A.S., Tsang, Y., Kwak, J.T., & Rajpoot, N.M. (2018). Hover-Net: Simultaneous segmentation and classification of nuclei in multi-tissue histology images. Medical image analysis, 58, 101563 .
> >
> > We have also reworded some parts of the abstract and introduction to better reflect the focus of this paper.

---

> > > ### Comment · Reviewer_KHyf · 2024-03-23
> > > **Reviwer KHyf final decision**
> > >
> > > Thank you for the comprehensive response and the meticulous revisions made to the manuscript based on the feedback provided. Your detailed explanations and the additional information included, particularly concerning the pipeline enhancements, threshold values for model inference, and the foreground background estimation, have significantly clarified my initial concerns. Moreover, the added chapters and studies, such as the ablation study and the augmentation strategies, alongside the modifications for speed improvements, have contributed to a deeper understanding of your work's value and its advancements in the field.
> > >
> > > I appreciate the effort to clarify the distinctions between your pipeline and previously published models, and the emphasis on making the model more efficient and usable on whole slide images. The inclusion of supplementary chapters and the correction of errors further demonstrate your commitment to transparency and rigor in your research.
> > >
> > > The updates regarding the batch size for CellViT and the correction of previous oversights, like the enabling of automatic mixed precision during inference, are particularly noteworthy. These changes not only address my concerns but also showcase the robustness of your evaluation process.
> > >
> > > Considering the authors' familiarity with high performance computing environments, I propose a strategy to improve your computational workflow and address the challenge of requiring 1600GB of RAM. This high memory demand often leads to prolonged job initialization times and restricts your operations to a single, high-capacity computing node. My recommendation is to incorporate dependency graphs and automated job arrays that operate independently. This approach would utilize OpenSlide's capability for delayed reading, allowing for data to be loaded into RAM only as necessary. Specifically, it would enable the loading of particular tile sets on demand, rather than the entire slides, ensuring that only sufficient tiles are available to maintain GPU activity during inference through the data loader. By adopting delayed reading, you could dynamically manage memory usage, thus reducing the need for extensive hardware and increasing the flexibility of job scheduling across various computing resources. Implementing this modification could significantly streamline your computational processes, making them more versatile and user-friendly within different high-performance computing contexts.
> > >
> > > Given these substantial revisions and clarifications, I am convinced of the manuscript's significant contributions to the field. Therefore, I have decided to change my decision from borderline to strong accept. The advancements you've made in developing a more efficient pipeline for whole slide image analysis, as well as the thoroughness of your experimental validation, are commendable. I look forward to seeing the impact of your work on future research and applications in the domain.

---

### Official Review · Reviewer_uqnd · 2024-03-05

**Confidence:** 4
**Preliminary Rating:** 5
**Recommendation:** Oral
**Final Rating:** 5

**Summary:**

The paper introduces a segmentation- and classification-based model for use in histopathology, specifically focused on H&E-stained images. The paper shows the evolution to the current iteration of the pipeline with the updated model HoVer-NeXt. The pipeline is particularly focused on H&E, given it is the most cost-effective of the staining technologies. The authors suggest the pipeline can be used for hypothesis generation. The pipeline performs favorably to previous works, being fast and accurate. The publication also offers an extension to the existing dataset, Lizard, for mitosis and eosinophils.

**Strengths:**

-	Overall, a well-articulated, comprehensive, and impactful paper. The transition from previous iterations to the current HoVer-NeXt illustrates a nice evolution over time. Extensive explanation of the background.
-	The paper offers several contributions, including: open source code, an update to existing open source dataset (i.e., Lizard), an updated model (i.e., HoVer-NeXt), and a pipeline that is faster than other existing WSI-geared pipelines.
-	Supports various Bio-Formats (through the use of OpenSlide).
-	The paper shows an extension from prior submissions, with the current model updated with ConvNeXt-v2 encoder for improved performance, class-based weighting removed to only have the standard focal loss, and the removal of the convex-hull post-processing to improve computational efficiency.
-	Comprehensive evaluation and comparison between CoNiC submission, then HoVer-NeXt models Tiny, Base and Large across different cell types.
-	Comprehensive qualitative evaluation in addition to quantitative assessment.
-	Clearly highlights why Hausdorff distances are higher (i.e., removal of convex-hull post-processing leading to more segmentation outliers).
-	Clear discussion on limitations of the work.

**Weaknesses:**

-	While the authors note that generating other data, such as immunofluorescence, is considerably expensive (and suggest their pipeline can be used on H&E for hypothesis generation), the adaption of this pipeline to immunofluorescence would be interesting and very useful. Particularly for those who work in spatial transcriptomics.
- Additionally, I would be interested to see extensions (given the focus here has been on H&E) to other staining techniques, such as CD31 or CA9.

**Detailed Comments:**

The paper is comprehensive, articulate, extensively acknowledges prior work, and is transparent about the pros and cons of the current pipeline. The major concern I have is to create a version (eventually) for immunofluorescence or staining markers for greater adaptability. But the authors address this point nicely in stating this can be used for hypothesis generation (given H&E is the cheapest and baseline staining technique).

**Justification Of Final Rating:**

The authors have produced a robust body of work and provided comprehensive responses to the review set out to them, including answering questions related to future modifications. I maintain my original score of a strong accept.

**Justification Of The Preliminary Rating:**

A very comprehensive piece of work that extensively highlights prior work up until this point, clearly outlines the evolution of the model in this paper, offers extensions to existing databases, provides open source code, and has extensive qualitative and quantitative assessments.

**Questions To Address In The Rebuttal:**

The biggest item to address would be future plans for this work and adapting it to other staining techniques.

**Special Issue:**

No

---

> ### Author Response · Authors · 2024-03-17
>
> We would like to thank the reviewer for taking the time to read and review our manuscript as well as for the follow-up questions, which we address below.
>
> We believe switching to a semi-supervised training regimen will increase options to switch to other modalities such as immunohistochemistry or immunofluorescence images. Our model currently requires many accurate manual annotations and is focused on classification, which in the context of spatial transcriptomics might be interesting but should always be extendable to give all information and not just the reduction to a single class. However, replacing the model and training process is rather straightforward and there is an ongoing effort in our lab to adapt the pipeline to spatial transcriptomic data. Regarding CD31 and CA9 specifically, as both are membrane markers and not located near the nucleus, the challenge here will be to correctly attribute the stain to the right cell. We are positive that this can be easily solved with labelled data using our pipeline, but at the same time also believe that a carefully engineered method using just nuclei instance segmentation methods will be sufficient.

---

> > ### Comment · Reviewer_uqnd · 2024-03-28
> >
> > I thank the authors for their comprehensive response to myself and all reviewers. I echo their sentiments in complimenting the authors for producing a robust body of work. The proposed suggestions to augment the capabilities of the existing pipeline (e.g., semi-supervised training to adapt to alternate modalities, such as immunofluorescence) and the ongoing efforts to adapt the pipeline for spatial transcriptomics are implementations I am looking forward to seeing in the future. I maintain my original score of a strong accept.

---

### Author Response · Authors · 2024-03-17

Dear Review committee,

We are grateful for the opportunity to submit a revised version of our manuscript entitled “HoVer-NeXt: A Fast Nuclei Segmentation and Classification Pipeline for Next Generation Histopathology”. We would like to thank the reviewers for their comments and feedback, which we feel have now significantly improved the paper. Please find our each of our replies in a point-by-point manner below, where we address each comment. All changes made to the manuscript have also been indicated in the rebuttal. All changes in the manuscript have been highlighted in teal. We hope that you find this revision satisfactory and that our work will be of great interest to the readers and attendees at MIDL 2024.

---

### Meta-Review · Area_Chair_81Hh · 2024-03-30

**Recommendation:** Accept (Oral)
**Confidence:** 5

**Metareview:**

The work introduces HoVer-NeXt, an advanced model for nuclei segmentation and classification in histopathology, improving upon a previous challenge submission. It offers extensions, simplifications, and new training/validation data, validated on the PanNuke dataset. The model, focusing on H&E-stained images, is efficient for WSI analysis, promoting hypothesis generation and offering competitive speed and accuracy. An expanded dataset, Lizard, is provided for mitosis and eosinophils study. HoVer-NeXt, with varying sizes, and its source code are publicly available on GitHub, under GPL-3.0, enhancing its application in biomedical analyses.

The paper is notable for its comprehensive contributions to biomedical research, particularly through its commitment to openness and transparency by providing all code and data necessary for model training and inference. Its development of an efficient WSI inference pipeline, which outperforms existing methods in both time savings and performance, is a significant strength. The clarity and accessibility of the manuscript enhance its impact, with resources including ConvNextV2 weights being made publicly available, fostering reproducibility and further research. Innovations such as the updated HoVer-NeXt model and a faster processing pipeline, alongside extensive evaluation and methodological refinements, underscore the paper's robustness and its forward leap in computational efficiency and accuracy in the field of biomedical imaging.

Reviewers provide unanimously positive ratings.

---

### Decision · Program_Chairs · 2024-04-05

Accept (Oral)